

# Preservation of latest Cretaceous (Maastrichtian)—Paleocene frogs (*Eorubeta nevadensis*) of the Sheep Pass Formation of east-central Nevada and implications for paleogeography of the Nevadaplano

Joshua W. Bonde[1], Peter A. Druschke[2], Richard P. Hilton[3], Amy C. Henrici[4] and Stephen M. Rowland[5]

[1] Department of Conservation and Research, Las Vegas Natural History Museum, Las Vegas, NV, United States of America
[2] ExxonMobil Upstream Oil and Gas, Houston, TX, United States of America
[3] Natural History Museum, Sierra College, Rocklin, CA, United States of America
[4] Section of Vertebrate Paleontology, Carnegie Museum of Natural History, Pittsburgh, PA, United States of America
[5] Department of Geoscience, University of Nevada—Las Vegas, Las Vegas, NV, United States of America

Corresponding author
Joshua W. Bonde, paleo@lvnhm.org

## ABSTRACT

Here we report on exceptional preservation of remains of the frog *Eorubeta nevadensis* in deposits of the Sheep Pass Formation, ranging from Late Cretaceous to Eocene, in the south Egan Range, Nevada. This formation represents a lacustrine basin within the Sevier retroarc hinterland. The formation is subdivided into six members (A–F); of interest here are members B and C. The base of member B is ?uppermost Cretaceous-Paleocene, while member C is Paleocene. Member B frogs are preserved in three taphonomic modes. Mode 1 frogs are nearly complete and accumulated under attritional processes, with frogs settling on microbial mats, as evidenced by crenulated fabric of entombing limestone. Mode 2 involves accumulation of frogs as a result of attritional processes. These frogs are mostly complete with some showing evidence of invertebrate scavenging. Possible scavengers are gastropods, ostracods, and decapods. Mode 3 is represented by isolated, reworked remains of frogs as a result of storm activity, supported by the association of elements with disarticulated bivalves and mud rip-up clasts. Member C preserves frogs in two taphonomic modes. Mode 4 are nearly complete frogs that accumulated during discrete mass mortality events. Numerous individuals are preserved along bedding planes in identical preservational states. Mode 5 is beds of frog bone hash, which represent increased energy to the depositional system (likely tempestites) and reworking of previously buried frog remains. Taphonomy of the frogs, along with the associated fauna and flora, are consistent with preservation in a cool, temperate lake basin, supporting previous interpretations that the Nevadaplano was an elevated plateau during the late Cretaceous through the Eocene. This is a period of time coincident with a climatic thermal optimum, thus the most parsimonious explanation for a temperate lake at the latitude of east-central Nevada is to invoke high elevation, which is consistent with independent structural and clumped stable isotope studies.

## INTRODUCTION

Upper Cretaceous to lower Paleogene deposits of the Sevier retroarc foreland basin system of the western United States have yielded a wealth of fossil plant, invertebrate and vertebrate remains; however, comparatively few paleontological studies exist for coeval intermontane deposits of the Sevier retroarc hinterland region. The Upper Cretaceous-Eocene Sheep Pass Formation of east-central Nevada represents deposits of a synconvergent extensional basin within the Sevier retroarc hinterland (*Druschke, Hanson & Wells, 2009*; *Druschke et al., 2009*; *Druschke et al., 2011*). It occupied what is widely interpreted as a high-elevation orogenic plateau (*Coney & Harms, 1984*; *Jones, Sonder & Unruh, 1998*; *Dilek & Moores, 1999*; *DeCelles, 2004*; *Snell et al., 2014*). Previous studies of the Sheep Pass Formation type section have focused on palynology (*Fouch, 1979*) and invertebrates such as mollusks (*Good, 1987*) and ostracodes (*Swain, 1987*). Remarkably, the only fossil vertebrate that is known from the >1 km thick Sheep Pass Formation type section are the fossil frogs discussed herein, which recently were identified (*Henrici et al., 2018*) as *Eorubeta nevadensis*. Ancient high-elevation sedimentary packages are rarely preserved in the stratigraphic record due to intense erosional processes at high elevations over extended periods of time. In addition to the processes of erosion, the Sevier hinterland has been subjected to several episodes of Paleogene and Neogene extension and volcanism that have further disrupted the original continuity of synorogenic deposits of the Sheep Pass Formation (*Druschke, Hanson & Wells, 2009*; *Druschke et al., 2009*; *Druschke et al., 2011*). Given the rarity of ancient high-elevation sedimentary deposits, it is even rarer to find records of ancient high-elevation biotas. Thus the Sheep Pass Formation provides a unique opportunity to investigate the preservation of biological remains in a high-elevation setting.

During the course of this study, we have recovered dozens of partial to nearly complete fossil frog specimens (34 frogs from Member B and 45 from Member C, not counting those that are fragmentary) from the Sheep Pass Formation type section, many of which are mostly complete, as well as a number of laterally extensive frog bonebeds. Exceptional preservation in this case refers to common preservation of nearly complete individuals as well as the abundance of elements; no soft tissue preservation has been found as yet. Frogs are the only vertebrates identified within the Sheep Pass Formation type section to date (except for one fragmentary mandible which may be a mammal as well as several other as yet unidentified large isolated fragmentary bones). We have also documented potential decapods, trace fossils, and scattered plant remains, in association with previously documented ostracod and mollusk faunas. These discoveries present an interesting pattern of community structure and preservation within the Sevier hinterland during the latest Cretaceous and Paleogene. In this article we describe the taphonomic parameters of fossil material collected from the Sheep Pass Formation type section and interpret the

preservational modes of the fossils. We also consider what this fossil material may reveal about the ecology and evolution of this long-lived, high-altitude lake basin.

## Geologic setting

The Sevier orogen is typified by thin-skinned thrust faulting and resultant crustal thickening in the Sierra Nevada retroarc region, resulting from prolonged Jurassic to Paleogene eastward subduction of the oceanic Farallon plate beneath the western margin of North America (*DeCelles, 2004*, and references therein). Following maximum crustal thickening in the Late Cretaceous, east-central Nevada is generally envisioned as part of a high-elevation orogenic plateau (*Coney & Harms, 1984*; *Jones, Sonder & Unruh, 1998*; *Dilek & Moores, 1999*; *DeCelles, 2004*). Synconvergent, surface-breaking normal faults documented within the Sheep Pass Formation suggest that syncontractional extension had initiated by latest Cretaceous time in the Sevier hinterland of east-central Nevada, resulting in a series of basins generally analogous to the high-elevation graben systems of the modern Puna-Altiplano and Tibetan Plateau (*Druschke, 2008*; *Druschke, Hanson & Wells, 2009*; *Druschke et al., 2009*). In support of a high-elevation interpretation, clumped stable-isotope analyses of lacustrine carbonates within the basal Sheep Pass Formation suggest a 2.2 to 3.1 km paleoelevation for east-central Nevada during the latest Cretaceous and earliest Paleocene, some 2.2 km higher than the foreland basin in Utah at that time (*Snell et al., 2014*). The Sevier hinterland was subsequently affected by a southward younging sweep of middle to late Eocene extension and associated volcanism (*Armstrong & Ward, 1991*; *Gans et al., 2001*) that reactivated elements of the Sheep Pass basin system (*Druschke, Hanson & Wells, 2009*). Most recently, large-magnitude Neogene Basin and Range extension subjected the Sheep Pass Formation to differential uplift, erosion, and burial beneath younger extensional basins.

The Sheep Pass Formation, first described by *Winfrey Jr (1958)* and *Winfrey Jr (1960)*, is a sedimentary package that forms isolated outcrops in various mountain ranges of east-central Nevada (Fig. 1). The Sheep Pass Formation is divided into members A-F based largely upon lithology (Fig. 2). Previous workers have determined that the primary depositional settings of the Sheep Pass Formation are lacustrine, alluvial fan, and fluvial (*Winfrey Jr, 1958*; *Winfrey Jr, 1960*; *Kellogg, 1964*; *Fouch, 1979*; *Druschke, 2008*).

Whereas the current study documents the first identifiable vertebrate remains from the type section, previous studies have recorded vertebrate remains from other localities within the Sheep Pass Formation. *Fouch (1979)* identified the remains of the insectivore-like mammal *Nyctitherium* within Paleocene to Eocene lacustrine carbonates of the Grant Range (Fig. 1). *Emry (1990)* identified a mammalian fossil assemblage, including also anurans, lizards, and snakes of Eocene (Bridgerian) age at the northern Egan Range, Elderberry Canyon location. Of most direct relevance to the current study, *Hecht (1960)* identified the remains of two fossil frogs recovered from a petroleum exploratory drill core located just west of the Sheep Pass Formation type section in White River Valley. These specimens, which he assigned to a new genus and species, *Eorubeta nevadensis*, were recovered from a lacustrine limestone correlative to Member B or C. *Hecht (1960)* noted that frog population densities would have to have been very high to preserve the two specimens found within a

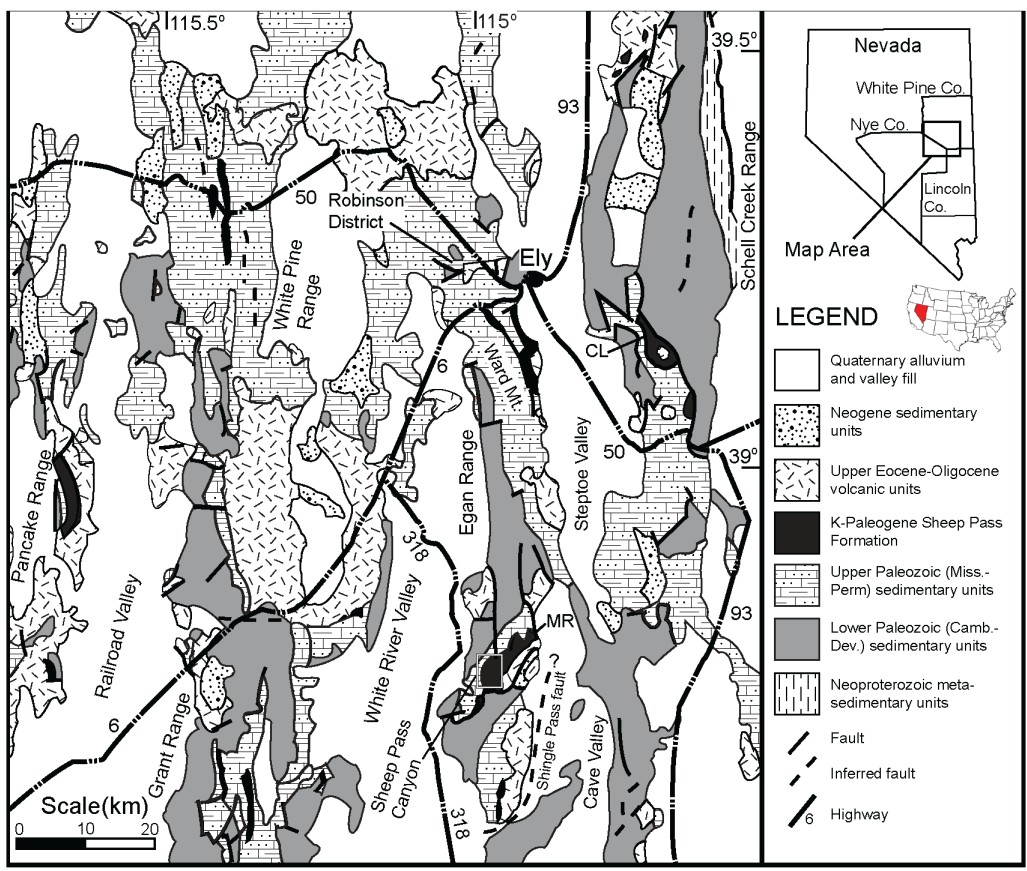

**Figure 1** **A geologic map of east-central Nevada.** The Sheep Pass Formation type-section is denoted in the small white inset box in the south Egan Range.

single core, although it would take nearly fifty years for specimens to be found in natural outcrops within the Sheep Pass Formation type section.

Absolute age control places the maximum depositional age of the basal Member A at 81.3 ± 3.7 Ma based upon (U-Th)/He cooling ages of detrital zircons, with the upper portion of Member A also yielding two euhedral detrital zircons with respective U-Pb ages of 68 and 70 ± 1 Ma (*Druschke et al., 2009*). The basal part of Member B in the type area is 66.1 ± 5.1 Ma based upon U/Pb dating of biogenic carbonates (*Druschke et al., 2009*). The rest of Members B and C is constrained as Paleogene based upon biostratigraphy (*Fouch, 1979*; *Good, 1987*; *Henrici et al., 2018*). Taking into account the inherent uncertainties in previous absolute and biostratigraphic age control, the K-Pg boundary may be present in the Sheep Pass type section, but to date this has not been identified.

# MATERIALS & METHODS

## Prospecting and surface collection

Fossil localities were discovered by prospecting exposures of the Sheep Pass Formation type section within the Sheep Pass Canyon area. The positions of fossil sites, including isolated

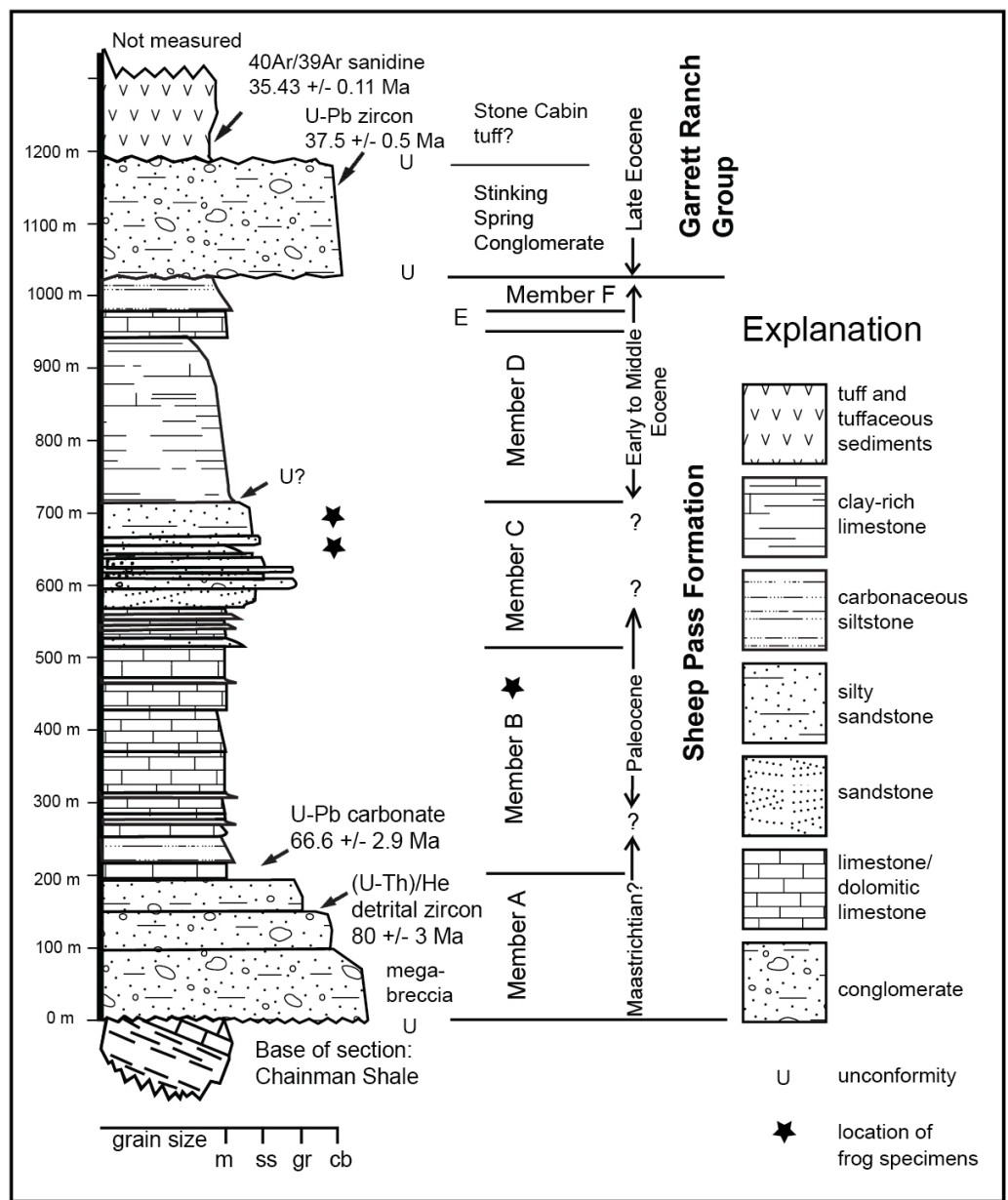

**Figure 2 Stratigraphy of the Sheep Pass Formation type section showing where absolute dates have been aquired and approximate distribution of frog fossils.** Frogs are found throughout Member B, and in discrete beds at the top of Member C.

elements, were plotted on a topographic map, their GPS coordinates recorded (these coordinates are on file at the respective repositories of the specimens), then specimens were taphonomically assessed before being collected (those reposited at Sierra College Natural History Museum and Las Vegas Natural History Museum) or later in the laboratory (those reposited at Carnegie Museum of Natural History).

Sedimentological data were recorded at each fossil locality to determine their depositional environments. Only surface collections were made; no excavations were conducted. Fossil material was noted as float or as *in situ*. *In situ* material was discovered by splitting the exposed mudstones with a rock hammer. Collected specimens are reposited at:

Sierra College Natural History Museum, Rocklin, CA (SCNHM)
Carnegie Museum of Natural History, Pittsburgh, PA (CM)
Las Vegas Natural History Museum, Las Vegas, NV (LVNHM)
(all figured specimens in this paper are housed at the SCNHM or the CM)

Each fossil locality was placed within the stratigraphic framework of *Druschke (2008)*, which allowed for identification of fossiliferous members of the Sheep Pass Formation. All fossils so far discovered are from Member B (?Upper Cretaceous-Paleocene) and Member C (Paleocene). Locality data are reposited with specimens at their respective institutions. All field work was conducted under permit from the Bureau of Land Management (8270(NV040) 2009 to RPH).

## Taphonomic analysis

Taphonomic observations recorded included spatial distributions of specimens, degree of articulation and/or close association of disarticulated elements, presence of bone-on-bone contacts, types of bone breakages, and alteration halos. These taphonomic modifiers were revisited in the lab as well, and under light microscope. We obtained data on surface modification in the lab, after preparation of elements, because such features are often obscured by matrix (*Eberth, Rogers & Fiorillo, 2007*). Surface modification data include weathering (after *Behrensmeyer, 1978*), abrasion (after *Shipman, 1981*), tooth marks, bioerosion, trample marks, and the nature of breaks (e.g., faults, blocky, spiral). Trend and plunge data of long-bone elements were measured to determine whether elements have been aligned due to fluid flow, or whether the bones are oriented randomly.

## MEMBER B PALEONTOLOGY & TAPHONOMY

Member B of the Sheep Pass Formation type section has produced a diverse assemblage of plants, invertebrates, and vertebrates. Ostracods are the most numerous invertebrates, and frogs are the only vertebrates recovered to date, with the exception of a toothless mammal jaw. In addition to body fossils, there are also some invertebrate traces preserved within Member B. Plant body fossils are present in isolated horizons throughout Member B. Fossils are found in at least three different lithofacies: dolomitic clayshales, dolomitic microbialites, and tempestites.

### Plants

There are at least four types of plant body impressions. The first type consists of portions or bracts of a larger organ that is 1 cm long and up to 1.5 mm in diameter. Perpendicular to the long axis are shorter appendages. The second type consists of long (up to 20 cm), 2–3 cm wide impressions, with parallel structures along the long axis. These impressions maintain their width from the base until they finally taper to a tip at their distal end (Fig. 3). These are the best preserved plant specimens. Both types of plant fossils are found in a very

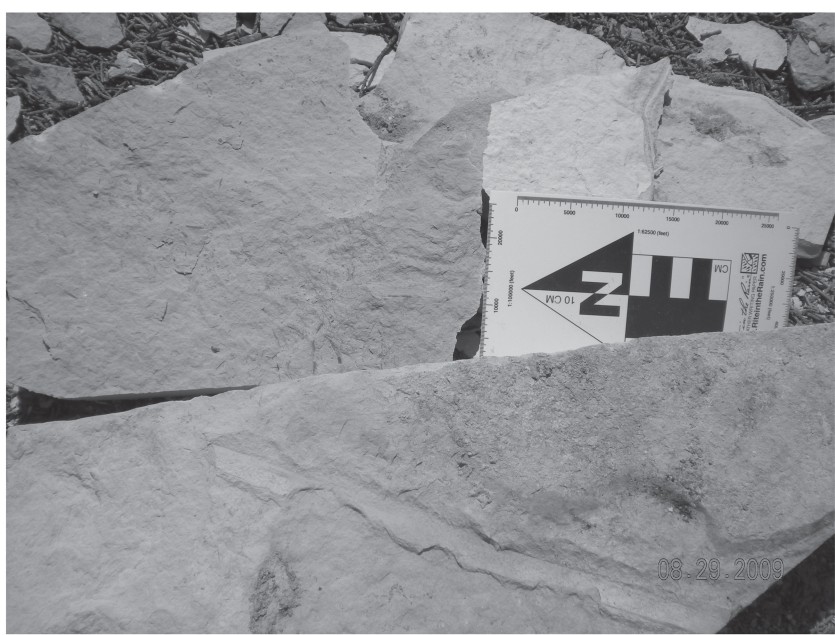

**Figure 3 Representative plant fossils.** Plant impressions along a fine-grained mudrock horizon from Member B.

fine dolomitic clayshale, which contains only plant fossils; no animal body or trace fossils are present. The third type is a single, unidentified, angiosperm leaf, roughly 5–6 cm from petiole to the tip of the leaf (Fig. 4). A dominant primary vein extends from the petiole to the tip of the leaf, with arcuate secondary veins occurring on opposite sides of the primary vein. The margin of this leaf is entire and is eucamptodromous in morphology (Fig. 4) (cf. *Hickey, 1973*). This specimen was found in an irregularly laminated, dolomitic mudstone. The fourth type is a partial gymnosperm leaf. It is 2–3 cm wide, with the distal portion not preserved . The venation is dichotomous, with an entire margin and acute base, consistent with a ginkgophyte (*Tidwell, 1998*).

Type 2 impressions are likely autochthonous and represent a period of time when the lake level was low enough for these plants to subsist in this portion of the basin. Lower and higher in the stratigraphic section the water was probably too deep for such plants. This interpretation is supported by the absence of fossil gymnosperms higher or lower in the section. Allochthonous macrofloral material in lakes rarely travels farther than 50 m from the source plant (*Ferguson, 1985*); thus we interpret the shore of the paleolake to likely have been within 50 m of the sites where these fossils were deposited.

The scarcity of plant fossils does not allow for a more comprehensive picture of the plant life within the basin in which the Sheep Pass Formation type section was deposited. Little can be said about the plant record, as to what the surrounding foliage was like, nor can these fossils lend themselves to rigorous paleoclimatic analysis. The previous palynological study by *Fouch (1979)* found mostly charophytes and algae.

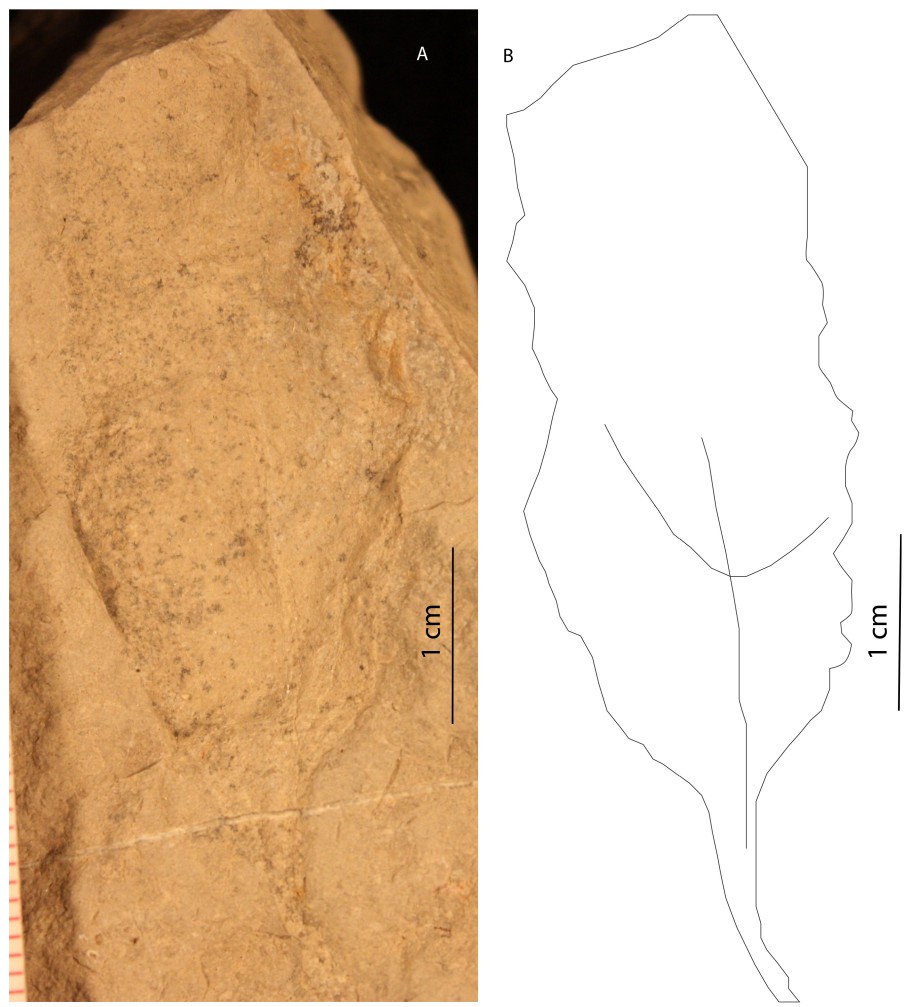

**Figure 4  Eucamptodromus leaf impression in Member B.** A eucamptodromus leaf impression in fine-grained facies of Member B. The line diagram illustrates the veination of the impression, showing the arcuate secondary veins.

## Invertebrates

Numerous invertebrate fossils occur in Member B, including body fossils of mollusks and crustaceans. Bivalves, typically less than a centimeter in diameter, are found isolated in planar-laminated, dolomitic mudrock lithofacies, or they occur concentrated in irregularly bedded dolomitic mudstone, along with fine-grained mudstone intraclasts. Ostracods are extremely abundant in planar laminated, dolomitic mudrock lithofacies. In some of these beds ostracods are the only fossils preserved in abundance, whereas in other beds they occur in close association with vertebrate remains. Ostracods present in this unit are *Clinocypris?*, sp.*Paracypridopsis?* sp., and *Cypridea bicostata* (*Swain, 1987*).

Occasional impressions of potential decapod carapaces are found within the dolomitic mudstone lithofacies, in both the planar-laminated and crenulated fabric beds. The impressions are typically partitioned into 4–5 segments (Fig. 5). These attached segments

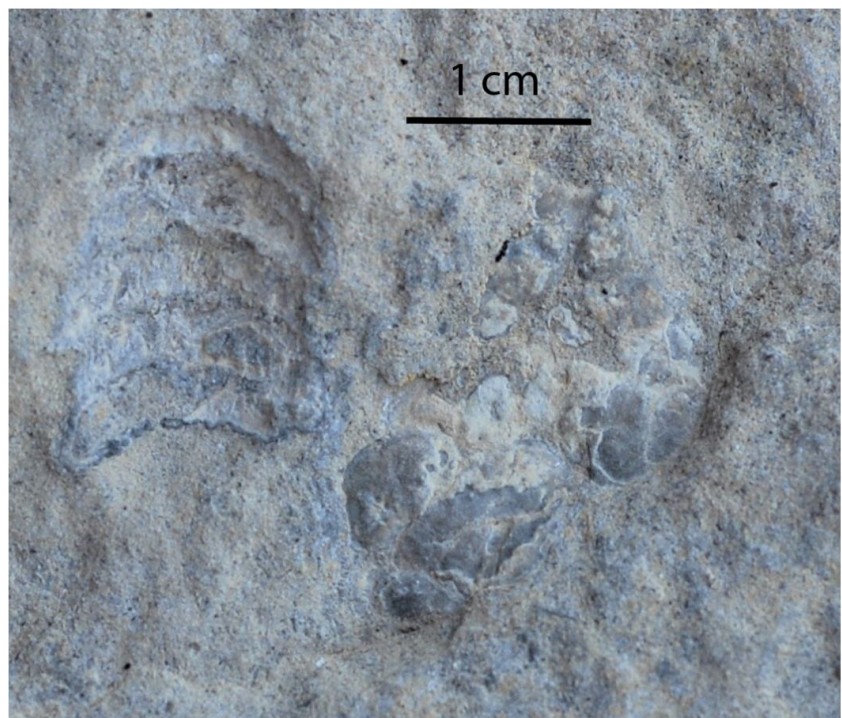

**Figure 5 Decapod molted exoskeleton.** A likely molted decapod exoskeleton. To the left is the telson with abdominal somites, to the right would be the cephalothorax.

are all slightly concave, in which each segment is nearly a centimeter in width and a few millimeters in length. The specimen shown in Fig. 5 is articulated with a more massive structure divided along a midline and oriented at 90° to the former structure. We interpret these structures to be articulated abdominal somites, part of the telson and posterior, dorsal parts of the cephalothorax of a potential decapod (similar to *Fetzner, 2002*).

Invertebrate remains and traces in Member B are typically found in a single lithofacies, the planar-laminated, dolomitic limestone. Ostracods tend to be concentrated, with dozens of individuals in a small area (less than a meter square). Previous workers concluded that ostracods (*Swain, 1987*) and mollusks (*Good, 1987*) from Member B are indicative of an alkaline, open-lake environment. In some instances the ostracods are found in direct contact with vertebrate bones (Fig. 6). The densest accumulations of ostracods are found in and around the cranium and other bone elements of frog fossils. This suggests that either the ostracods accumulated after the soft tissue of the frog was already decomposed or, perhaps, that the ostracods swarmed the frog carcass to scavenge the carrion. Ostracod swarming is a diagnostic sign of scavenging in both the fossil and modern records (*Wilkinson et al., 2007*). In addition to being found in close association with vertebrate remains, ostracods are found scattered along bedding planes of the planar-laminated limestones. Another occurrence of ostracods in Member B is along the agglutinated edges and meniscate infilling laminae of vertical decapod burrows. These specimens may either be accumulations of ostracods at normal background rates, or they may be the remains of individuals preyed upon or

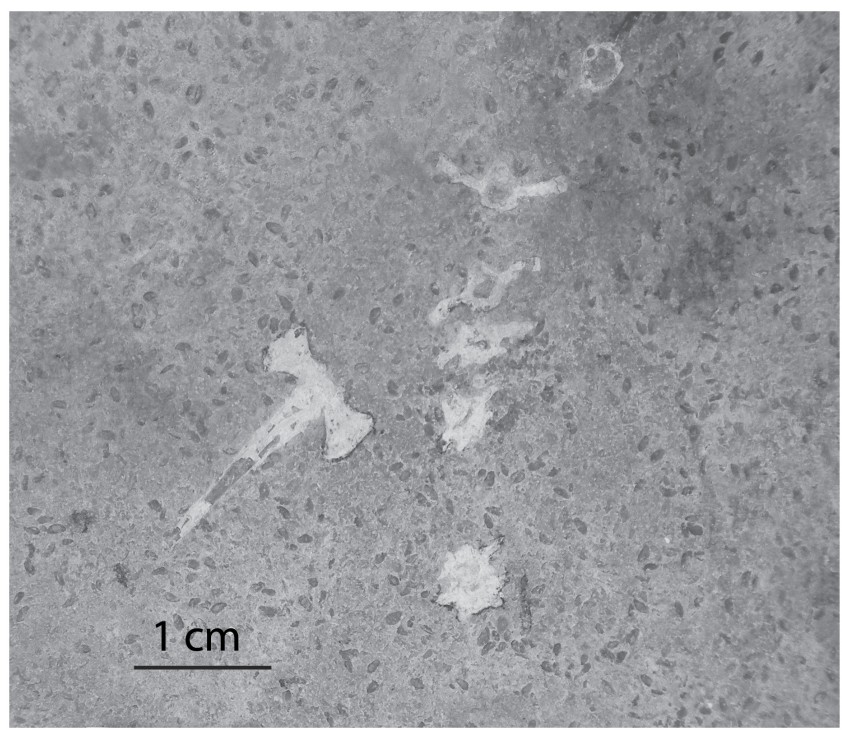

**Figure 6** **Ostracods in association with frog bones in Member B.** The laminated mudstone facies with a disarticulated frog vertebral column. Small oval structures are ostracod carapace remains.

scavenged by decapods. A study by *Gutierrez-Yurrita et al. (1998)* of crayfish gut contents showed that ostracods account for about 10% of crayfish diet. Ostracods are not found in any other lithofacies in Member B.

Other invertebrates preserved in Member B include occasional gastropods and numerous bivalves. *Good (1987)* identified this mollusk assemblage as a *Valvata, Hydrobia*-Sphaeriidae association; along with the ostracod assemblage, these mollusk taxa support the interpretation of an open, alkaline-lacustrine setting (*Good, 1987*). Analogous modern mollusk assemblages occur in temperate lakes of North America, in Ontario, Minnesota, and Maine (*Good, 1987*).

Bivalves are preserved in two taphonomic modes. The less common mode of preservation is as isolated, articulated specimens in planar-laminated, dolomitic limestone. These specimens are interpreted to be the result of attritional accumulation over time. The second, and more common, preservational mode is articulated and disarticulated valves found amongst irregular clasts of underlying clayshale strata in an otherwise silty, irregularly-bedded, dolomitic limestone. We interpret this preservational style to represent tempestite beds, with mudrock clasts being mud rip-up clasts. That, coupled with the dense accumulation of articulated and disarticulated specimens, suggests that at least some of the valves were remobilized. The irregular bedding, disarticulated valves, and mud rip-up clasts indicate that these beds represent periods of increased wave energy in the system

rather than fluvial influxes, because there are no signs of sedimentary structures indicative of fluvial processes, such as clast imbrications or cross-laminae.

## Vertebrates

Numerous frog specimens have been recovered from Member B of the Sheep Pass Formation within the Sheep Pass type section. These specimens are found stratigraphically throughout this member (Fig. 2). All specimens are preserved within dolomitic mudstones, and most of them exhibit dorsoventral compression.

Frogs are preserved in three taphonomic modes in Member B. Mode 1 includes individuals preserved as nearly complete skeletons only within those dolomitic mudstones with a crenulated-fabric (i.e., SCNHM VAF 3, Fig. 7). The lack of evidence of exposure and transport supports an interpretation that frogs in this taphonomic mode accumulated through attrition over time. This interpretation is supported by the vertical and lateral stratigraphic distribution of specimens that had settled onto microbial mats in life position and were subsequently buried and preserved; there are no discrete event beds. Crenulated laminations are an artifact of microbial mat sedimentation (*James, 1977*). Actualistic studies of fish carcasses reveal that water temperatures must be below 15 °C in order for a carcass to sink (*Elder, 1985*). In another actualistic study *Dodson (1973)* showed that a frog bloated and floating in a tank will begin to decompose and disarticulate after 21 days. In yet another actualistic experiment, frog carcasses began to decompose and disarticulate in just a few days (*Iniesto et al., 2017*). Similarly, the mostly complete frog specimens encountered in this study likely sank to the lake bottom within a few days of death. The lack of scavenging could be attributed to periods of anoxic or dysoxic conditions, as suggested by petroleum biomarkers from Sheep Pass Formation derived oil (*Ahdyar, 2011*). However, the relative abundance of invertebrate trace fossils within these crenulated fabric horizons suggests oxygen levels were not so low as to deter potential invertebrate scavengers from feeding on the lake bottom. A potential mechanism to inhibit scavenging in the presence of oxic waters is rapid carbonate precipitation mediated by the microbial mats (after *Dupraz et al., 2009*). Modern rates of carbonate precipitation in microbial settings can be as much as 100 μm per day (*Lebron & Suarez, 1996*). Rapid microbially mediated carbonate precipitation could have quickly removed frog carcasses from being accessible to invertebrate scavengers. Another mechanism which could have led to the intact preservation of frog carcasses is the initial entombment in a microbial mat sarcophagus (*Iniesto et al., 2017*). Such sarcophagus settings have been shown in a lab to preserve articulated frogs and soft tissue for up to three years (*Iniesto et al., 2017*), which could allow for sufficient time for burial in a lacustrine setting.

Taphonomic Mode 2 is similar to the first in that specimens are typically nearly complete; the difference is that there are no signs of microbial mats, as indicated by the lack of the crenulated fabric sedimentary structures. This mode is hosted in a planar-laminated, dolomitic limestone and, unlike the first vertebrate taphonomic mode, these are the horizons that contain abundant ostracods (cf. SCNHM VAF 26 A, Fig. 8). Many of the frogs preserved in this lithofacies are found in postures slightly more out of "life position" than those in the microbialites such as head to body angle or joint angles not in a relaxed

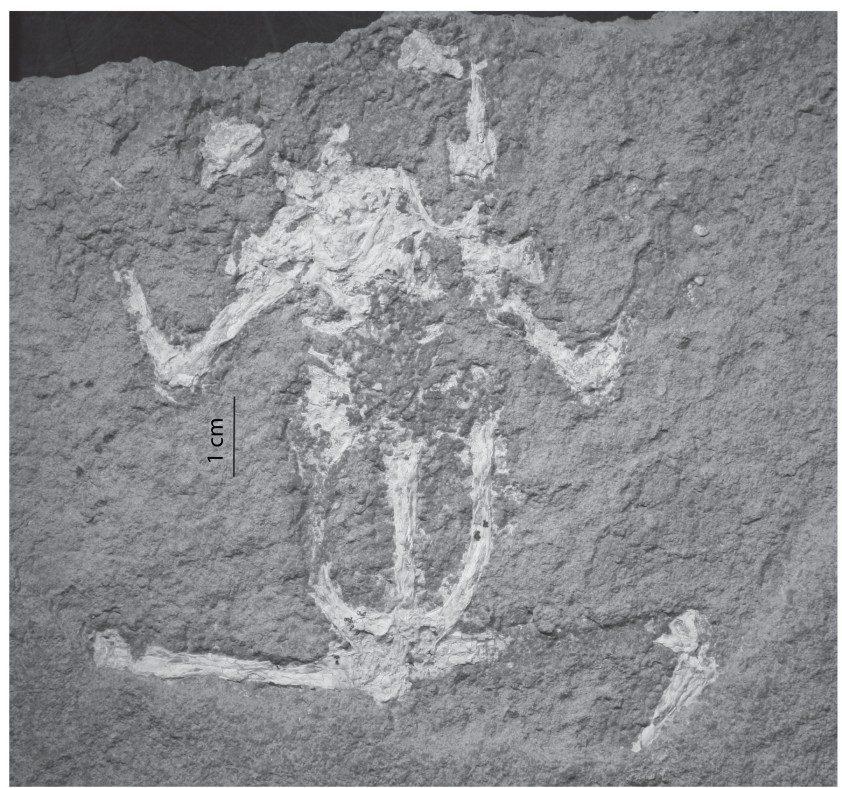

**Figure 7  Taphonomic Mode 1 of Member B.** A nearly complete frog (SCNHM VAF 3), in life position, preserved in the crenulated limestone fabric. This setting is interpreted as microbialite facies.

living frog posture, suggesting that they may have been subjected to additional transport or scavenging. As mentioned above, some of these frogs appear to have been scavenged by ostracod swarming. Other specimens with elements displaced in random directions from the main part of the animal are consistent with invertebrate scavenging (*Elder & Smith, 1984*), although minor lacustrine currents could have similarly displaced small elements. The fact that these frogs are also almost complete supports the interpretation that they must have sunk within three weeks of death if not much sooner (*Dodson, 1973*; *Iniesto et al., 2017*). In order to sink without bloating and disarticulating, water temperature was likely below 15 °C (*Elder, 1985*). In light of the stratigraphic distribution of this lithofacies, we interpret this taphonomic mode also to be attritional, with frogs dying and settling to the bottom through time.

The final taphonomic mode of frog elements in Member B, Taphonomic Mode 3, is related to the taphonomic mode described for bivalves in the beds interpreted as tempestite horizons. These frog elements are typically found isolated and associated with disarticulated bivalve valves and mud rip-up clasts. The frog bones are interpreted as having been reworked from either the lake bottom or underlying sediments. *Dodson (1973)* showed that even frog bones that have been submerged for long periods of time can be easily transported.

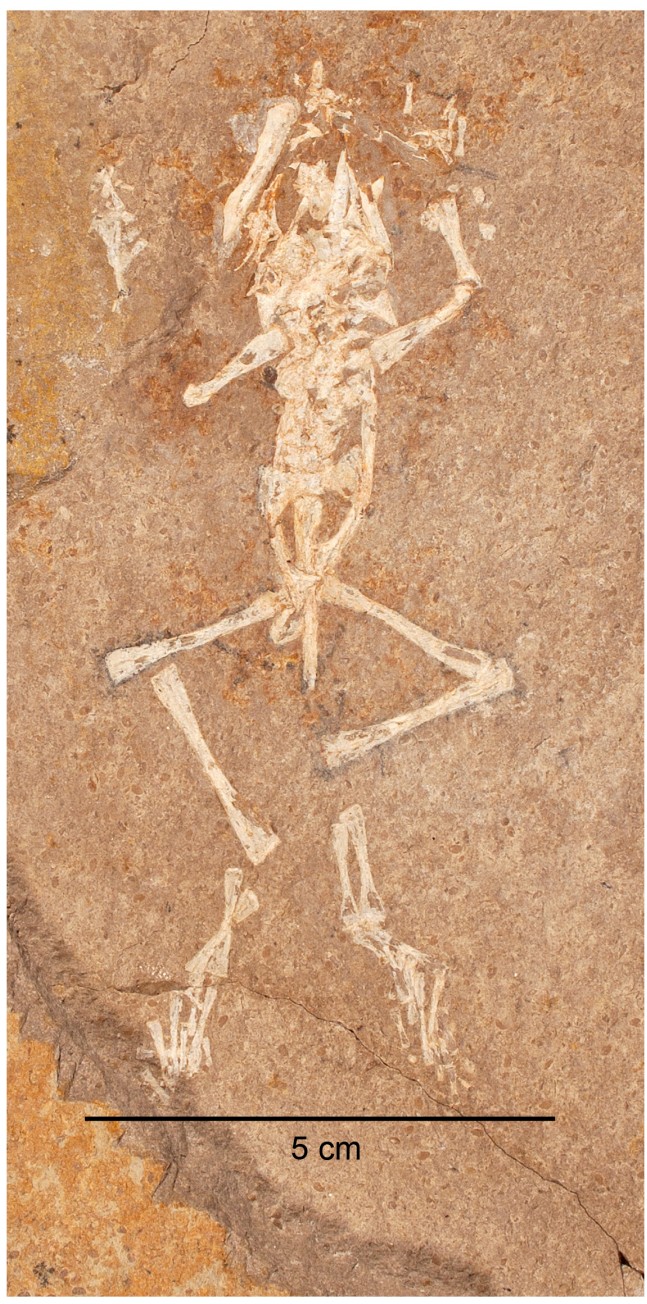

**Figure 8 Taphonomic Mode 2 of Member B.** A nearly completely articulated frog (SCNHM VAF 26), with some disarticulated and associated elements. This specimen also shows ostracod swarming, with carapace to bone contact in some places. This specimen is preserved in a laminated mudstone.

## Trace fossils

There are numerous burrows preserved within the dolomitic mudrocks of Member B. One type consists of randomly sinuous, continuous trails (Fig. 9), roughly 0.8–1.0 cm wide and U-shaped in cross section. These trails are observed only in irregularly laminated

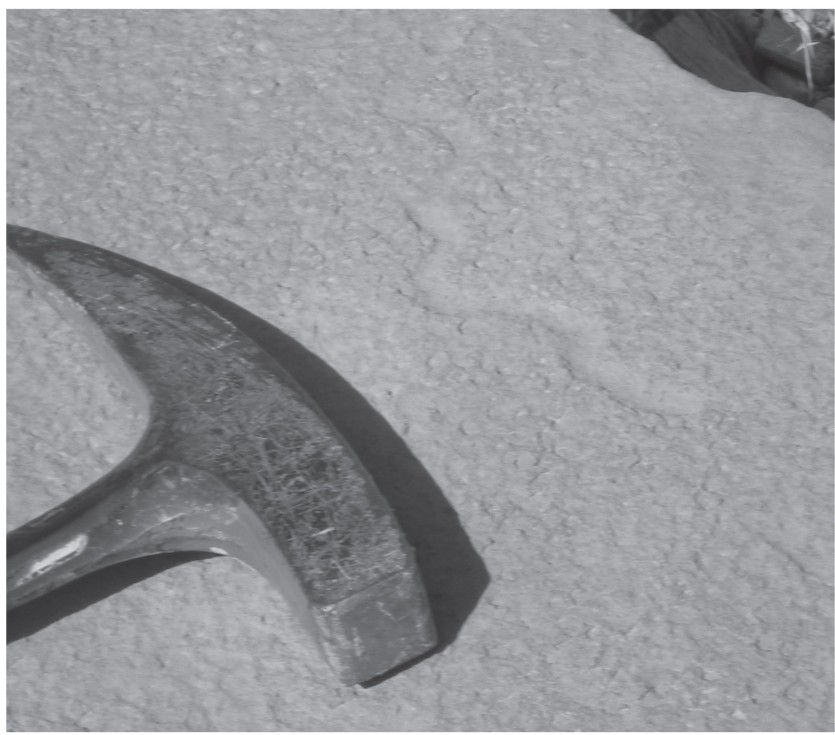

**Figure 9 Invertebrate trace through crenulated fabric of Member B.** An invertebrate trace fossil found in the crenulated, microbialite of Member B. Invertebrate activity shows that bottom waters were not anoxic or so dysoxic that invertebrate scavengers would not have had access to vertebrate remains.

mudrocks with crenulated fabric. Another common burrow stands out in slight positive relief, ~1 cm, along bedding planes of relatively flat laminated mudrock. In plan view, these are nearly perfectly circular, with diameters ranging from 5–7 cm, and they cut across stratigraphic layers to depths at least 6 cm deep. In the lab, we cut one of these burrows in half longitudinally to expose the internal structure (Fig. 10). Along the outer edge of the burrow, the mudrock becomes darker in color and is lined with occasional ostracods along the margins. In between the dark margins are very fine menisci of mudrock infilling the trace, with few ostracods. These burrows are found only in the ostracod-rich, horizontally laminated mudrocks, where they are common. These burrows match the characteristics of crayfish burrows in having, vertical orientation that crosses stratigraphy, and agglutinated mud margins (*Hasiotis, Kirkland & Callison, 1998*).

## MEMBER C PALEONTOLOGY & TAPHONOMY

A number of frog specimens have been recovered near the top of Member C (Fig. 2). In addition to anurans, this member preserves abundant mollusks, ostracods, and rare unidentified plant impressions. The most fossiliferous lithofacies in Member C are planar-laminated, silty limestones (packstones) and calcareous siltstones. Less common in the

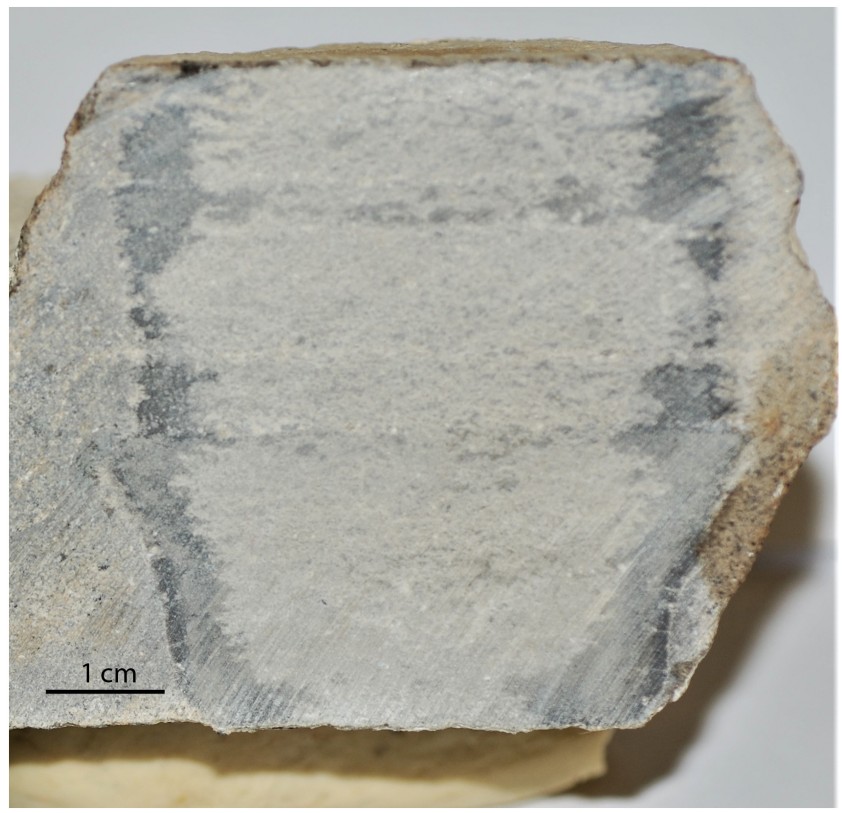

1 cm

**Figure 10  Sawed in half invertebrate burrow.** One of the circular invertebrate traces sawed in half, showing the internal structure. Notice along the edges the darker agglutinated periphery of the burrow.

member are trough-cross-bedded and ripple-marked litharenitic sandstones, trough-cross-bedded polymict conglomerates, and rare oncolitic limestones, but to date these lithofacies have not produced any vertebrate fossils.

Member C differs from Member B in that invertebrate remains are found within the same lithofacies and with similar preservation as the vertebrates. Two taphonomic modes are observed. Taphonomic Mode 4 consists of frogs, ostracods, and bivalves occurring together within horizontally laminated, calcareous siltstones (cf. SCNHM VAF 4, Fig. 11A) and in laminated to finely bedded silty ostracod packstones (cf. CM 89263, Fig. 11B). Ostracods in Member C do not show signs of "swarming" (in contrast to Member B), and bivalves are fully articulated, unlike those in the tempestite beds of Taphonomic Mode 3 in Member B. Ostracod density seems to be rather uniform, whereas bivalve density is variable. The depositional environment of Taphonomic Mode 4 is interpreted to be a lacustrine fan-delta (*Druschke, 2008*). As in the case of Member B, the ostracod and bivalve taxa indicate an alkaline pH (*Good, 1987*; *Swain, 1987*). Similar to Member B, frogs from this taphonomic mode are nearly complete and in life position. This suggests that the frogs were subjected to minimal or no dessication (*Smith, 1986*), transport, or scavenging. The well preserved "life posture" of these frogs is also similar to taphonomic processes occurring in Member B in requiring the body to settle to the bottom within

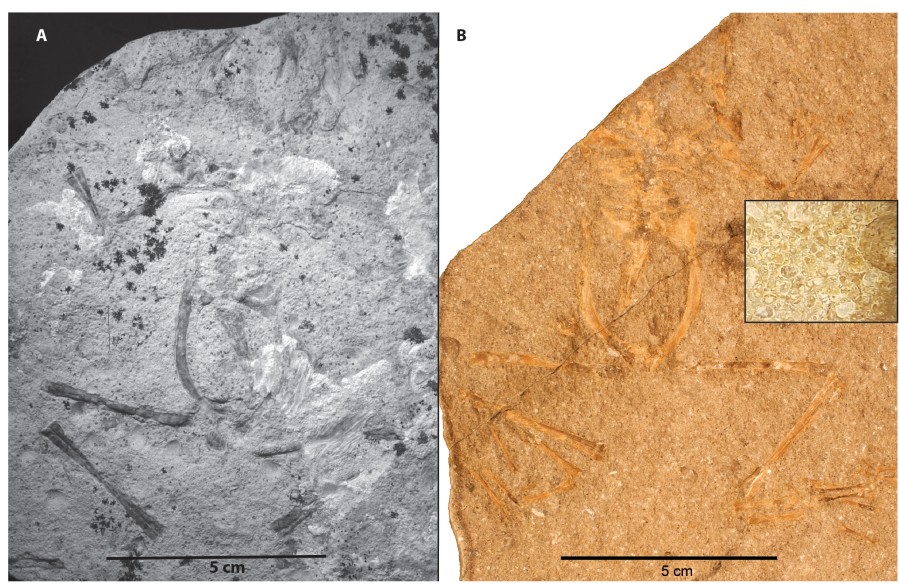

**Figure 11   Taphonomic Mode Four in Member C preserves frogs, ostracods and bivalves in near life position.** These frogs are preserved in silty, laminated limestones as in A (SCNHM VAF 4); or in ostracod-bivalve packstones as in B (CM 89263). The inset box in B shows a blown up view of the matrix, highlighting the packstone nature of the invertebrate remains.

a few days to three weeks of death (*Dodson, 1973*; *Iniesto et al., 2017*), and the low water temperature retarding bloating and disarticulation (*Elder, 1985*). Given that Member C is interpreted to be a deltaic environment, the sedimentary laminae are not varves. A low-energy environment is supported by the presence of articulated bivalves (*Good, 1987*). At least three horizons, and likely more as yet unidentified, of this taphonomic mode with multiple-individuals per bed suggests a recurring mortality event. There are no frog larval remains preserved in these beds, only post-metamorphic individuals of varying size. The similar preservational state and presence of more than one individual per bedding plane is further evidence that these were indeed discrete mortality events rather than attrition, in contrast to Member B Taphonomic Modes 1 and 2.

The accumulation mechanism may be a result of either biotic or abiotic factors. If these were terrestrial frogs, then the accumulation could be attributed to the congregation of frogs in the lake to breed. Abiotic accumulation mechanisms could have included carcasses being blown by lake fetch (e.g., *Henrici & Fiorillo, 1993*), low-density turbidity currents (*Smith, 1986*), or perhaps some other low energy current. Whether by biotic or abiotic factors, these individuals died in discrete events, were accumulated via biotic or abiotic factors, and sank to the bottom of the lake.

Taphonomic Mode 5 is odd, in that frogs are the only preserved organisms and they are completely disarticulated; invertebrates are completely lacking (cf. SCNHM VAF 11, Fig. 12). Given that none of the frog elements display evidence of subaerial weathering, we suggest that they were not exposed to air and light for any significant amount of time. Furthermore, no signs of abrasion are present, which supports the interpretation

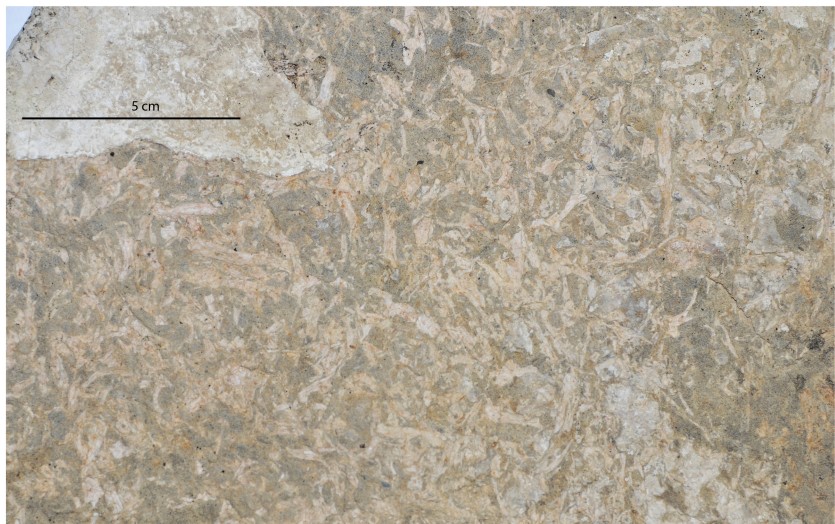

**Figure 12 Taphonomic Mode 5 preserved in Member C.** This specimen is a sample of the frog bonebed (SCNHM VAF 11). This horizon extends for over a kilometer laterally there, is no preferred orientation of elements, and the bones are preserved in bone to bone contact in a silty to very fine sandstone matrix.

of minimal transport distance. The bones show no strongly preferred orientation. The presence of bones of a wide array of sizes in a poorly-sorted matrix supports the conclusion that these elements were deposited in a single depositional event. An interesting observation is that many elements occur in a bone-to-bone contact relationship, indicating that the bones were denuded of flesh prior to burial. There are no observable articulations or close associations, suggesting that the animals were disarticulated when they became entrained within the depositional environment. These bones likely represent the reworked remains of individuals found at the base of these discrete bone-bearing beds of Taphonomic Mode 4 for Member C. There does not appear to be any winnowing of elements as per *Voorhies (1969)* groups. These bonebed horizons can be traced laterally over a kilometer.

Actualistic studies of frog and toad carcasses in pond water show that frogs can begin disarticulating within 21 days, but they remain mostly articulated for up to 45 days (*Dodson, 1973*). Therefore the depositional event was not likely the cause of death; the frogs were already dead and in the catchment before the event. There are no tell-tale signs of scavenging of elements, such as green-stick fractures or tooth marks. Mass die-offs of anurans are known to occur today (e.g., *Lips, 1999*; *Rachowicz et al., 2006*) and have been invoked in other prehistoric instances (*Henrici & Fiorillo, 1993*). Given the wide range of element sizes and random orientation of elements, we interpret this taphonomic mode to be a non-selective event assemblage. *Henrici & Fiorillo (1993)* proposed lake fetch as one of several hypotheses for the dense accumulation of frogs in a nearshore environment. We hypothesize that in this instance the frogs were already dead and decomposing, and that the concentrating mechanism was also sedimentary. In this case we cannot conclusively determine the depositional mechanism for the jumbled assemblage, perhaps a sediment gravity flow (as per *Smith, 1986*), or a tempestite bed similar to those of Member B.

## DISCUSSION

Results of the taphonomic analysis complements previous studies (*Coney & Harms, 1984*; *Jones, Sonder & Unruh, 1998*; *Dilek & Moores, 1999*; *DeCelles, 2004*; *Snell et al., 2014*), which indicate that the Sheep Pass Formation was deposited in a high-elevation setting. The preservational modes of the fossil frogs indicates that cool water temperatures prevented them from bloating and floating before settling onto the lake bottom. This seems surprising as periods of extreme global warming occurred during the early Paleogene (*Zachos et al., 2001*), which should have resulted in water temperatures warm enough to cause carcasses to bloat and float. High elevation of the Sheep Pass Formation lake system is the most likely explanation for cool water temperatures. Further support for the Sheep Pass basin being a cool-water lacustrine system is found in *Good (1987)*, who observed that, although palynological data from the Rocky Mountain region indicates a tropical to subtropical climate during the time of Sheep Pass Formation deposition, a modern analog for the Sheep Pass Formation molluscan association of *Valvata, Hydrobia*, Sphaeriridae in a tropical to subtropical realm does not exist. It was when *Good (1987)* looked to literature on more temperate, cooler water lakes in Ontario, Minnesota, and Maine that he found modern analogs to the Sheep Pass Formation molluscan fauna.

Although the frogs of taphonomic modes 1–4 are mostly complete, they show no signs of soft-tissue preservation, in contrast to other examples of frog-bearing lagerstatten. Although soft-tissue can be conserved for a number of years via microbial mat action (*Iniesto et al., 2017*), geological preservation requires subsequent sedimentary deposition. Soft-tissue preservation through geologic time can be mediated by anoxic conditions (*McNamara et al., 2012*). In the Sheep Pass Formation Taphonomic Mode 1, microbial mats likely helped to initially hold carcasses together, but through post-burial diagenesis any soft-tissue impressions or residues were probably lost. Post-burial diagenetic processes observed in Sheep Pass Formation rocks include petrogenesis and dolomitization; these chemical processes likely did not facilitate the preservation of soft-tissue post-deposition.

A puzzling aspect of the paleontology of the type section of the Sheep Pass Formation is that no identifiable remains of any vertebrates other than frogs, with the exception of the partial mammal jaw, have been found, including aquatic and semi-aquatic groups such as fish, turtles, crocodilians, or waterfowl. The invertebrate fauna is considerably more diverse. One locality within the Sheep Pass Formation at Elderberry Canyon, which lies north of the type section, does produce a large and varied fauna of frogs, reptiles, birds and at least 30 different types of mammals (*Emry & Korth, 1989*; *Emry, 1990*). The mammalian fauna indicates an early Bridgerian age (Eocene, ca. 50–46 Ma), which is younger than the frog bearing strata at the type section.

## CONCLUSIONS

The Sheep Pass Formation type section, spanning the Maastrichtian to the Eocene, represents a highland, cool water, alkaline, lacustrine setting on the Nevadaplano of east-central Nevada. This section preserves an abundant invertebrate fauna of gastropods, bivalves, and crustaceans. It also preserves exceptionally abundant, well preserved, nearly

**Table 1  Summary of lithofacies, fossils present, and depositional environments, with vertebrate taphonomic modes.**

| Lithofacies | Plants & Inverts | Frogs | Depo. Environment |
|---|---|---|---|
| **Member B** | | | |
| **Dolomitic clayshale** | Plant impressions, no animal body or trace fossils | No frogs | Shallow lacustrine with emergent vegetation |
| **Crenulated-fabric, irregularly-laminated, dolomitic mudstone** | Allochthonous leaves, common invertebrate trace fossils | Abundant, nearly complete frogs | Shallow, lacustrine, microbial mats **Taphonomic Mode 1** |
| **Planar-laminated, dolomitic mudstone** | Abundant invertebrate body and trace fossils. No plants | Abundant, nearly complete and associated frogs | Low-energy lacustrine **Taphonomic Mode 2** |
| **Irregularly bedded mudstone with common mud rip-up clasts** | Abundant articulated and disarticulated bivalves | Isolated elements | Tempestites **Taphonomic Mode 3** |
| **Member C** | | | |
| **Horizontally-laminated, calcareous siltstones and packstones** | Abundant, articulated bivalves and ostracods. No plants | Common, nearly complete frogs. | Lacustrine delta **Taphonomic Mode 4** |
| **Massive, irregular bed of varying grain-size and element density** | No plants or inverts | Abundant, disarticulated frog elements | Sediment gravity flow or tempestite **Taphonomic Mode 5** |

complete, semi-articulated, and close associations of elements of numerous frog skeletons of the species *Eorubeta nevadensis*. The majority of frog carcasses were preserved in an environment conducive to rapid settling to the lake bed without going through the bloat-and-float process. Accumulation processes differ between the two members in that frogs found in Member B are a result of attritional processes, whereas those in Member C were subjected to punctuated mortality events. These taphonomic modes (Table 1) should be useful in formulating a model of other high-elevation biotas and the preservation of biological remains into the rock record.

# ACKNOWLEDGEMENTS

We thank all those who helped with lab and field aspects of this project: Ty-Lor Birthisel, George Bromm, Tina Campbell, Frankie and Bob Jackson, Tom Madsen, and Pat McShea. We also thank Carrie Druschke, Becky Hall, Gene Hattori, Kristin Hilton, Jim Schmitt, Michael Wells, and our children for their support. Finally we thank the reviewers Miguel Iniesto, Dave Varricchio, an anonymous reviewer, and editor of this manuscript Fabien Knoll, as well as the reviewers of a previous version of this paper.

## Funding

The authors received no funding for this work.

## Competing Interests

Peter A. Druschke is employed by ExxonMobil Upstream Oil and Gas, his employer had no influence on the study design or interpretations.

## Author Contributions

- Joshua W. Bonde, Peter A. Druschke and Amy C. Henrici conceived and designed the experiments, analyzed the data, prepared figures and/or tables, authored or reviewed drafts of the paper, and approved the final draft.
- Richard P. Hilton and Stephen M. Rowland conceived and designed the experiments, analyzed the data, authored or reviewed drafts of the paper, and approved the final draft.

## Field Study Permissions

The following information was supplied relating to field study approvals (i.e., approving body and any reference numbers):

The Bureau of Land Management approved fieldwork [8270(NV040) 2009].

## Data Availability

A list of relevant repositories and accession numbers of specimens referred to in the article is available as a Supplemental File. SCNHM VAF 3, SCNHM VAF 4, SCNHM VAF 11, and SCNHM VAF 26 A are available at the Sierra College Natural History Museum. CM 89263 is available at the Carnegie Museum of Natural History.

## Supplemental Information

Supplemental information for this article can be found online at http://dx.doi.org/10.7717/peerj.9455#supplemental-information.

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
