# Peer review of "Preservation of latest Cretaceous (Maastrichtian)—Paleocene frogs (Eorubeta nevadensis) of the Sheep Pass Formation of east-central Nevada and implications for paleogeography of the Nevadaplano"

_PeerJ, doi:10.7717/peerj.9455_

## Round 0.1 · original submission · Major Revisions

You seem to have missed important references (see in particular Reviewer 1's comments). I would appreciate if you could provide better photographs for most of your figures.

Please, together with your unmarked revised manuscript, provide a marked-up copy as well as a document explaining how you have addressed each of the points raised by all three reviewers.

·

Basic reporting

Clear and unambiguous, professional English used throughout.
The article is well written and it is really easy to follow the reasoning all along. To the best of my knowledge, the English used is correct and should not be improved (the authors’ English is much better than mine)

Literature references, sufficient field background/context provided.
This is clearly one of the major weaknesses of the paper. The geological context is clear and well documented, but the relevant part of the paper, the study and description of frog preservation, is mainly based in 2 or 3 old references, which disregards the value of modern actuotaphonomic and observational papers. For instance, it is a surprise to read a paper on exceptional frog preservation that ignores great contributions to the field of taphonomy of frogs such as McNamara et al. PALAIOS, 27(2):63-77 or Báez Cret. Res. 41:90–106. Even if the geological periods or the chemistry of the lake are not the same, the paper would benefit of a comparative taphonomy with previous exceptional fossils. In addition, the authors support their findings on actualistic studies from 1973 and 1985 and they do not cite an actuotaphonomic study performed with frogs studying their preservation on microbial mats, which seems quite interesting in this context (Iniesto et al 2017 Scientific Reports 7:45160). Consequently, I think that literature has to be revised before publication.

Professional article structure, figures, tables. Raw data shared.
The structure of the paper and figures are appropriate and informative. I suggest the introduction of a more general map in Fig.1 in order to better locate the formation in USA. I commend the authors for Fig.2 which is a clear but very complete stratigraphic section.

Self-contained with relevant results to hypotheses.
The authors present a huge amount of information from a coherent body of work.

Experimental design

Original primary research within Aims and Scope of the journal.
This article fits into PeerJ’s aims and scope.

Research question well defined, relevant & meaningful. It is stated how research fills an identified knowledge gap.
The present paper present a high altitude Konservat-Lagerstätten which is quite rare, making the manuscript interesting. Through a combination of observational and geological observations they tried to reconstruct the ancient ecology of the system and the preservation context. Although not explicitly presented, their objectives and questions can be inferred from the text (lines 69-72).

Rigorous investigation performed to a high technical & ethical standard.
The ethical and rigorousness of the study cannot be questioned. However, to reach a high technical standard, the paper would benefit from the incorporation of several common techniques used in taphonomic studies such as electronic microscopy, or infrared or Raman spectroscopy. For instance, the identification of mats is based only crenulated laminations (line 249) and anoxic conditions are presented as a possible explanation for preservation (lines 254-257). However, the geochemical and mineralogical analysis of the sediment surrounding the fossils would incorporate strong support (or not) to these hypothesis. Although classical taphonomy is highly welcome and appreciated, in my opinion it does not stand without analytical information.
Moreover, several references that shows several discrepancies with the text should not be avoided and taken into account. For instance, line 350 states that frogs can remain articulated for 45 days according to Dodson (1973) but more recent studies showed that frogs can decay quite fast and show disarticulation in a few days (Iniesto et al 2017 Scientific Reports 7:45160). You are welcome to interpret that one result is more likely than other but an explanation is needed (for more examples, see below -section “speculation”-)

Methods described with sufficient detail & information to replicate.
The information is sufficient.

Validity of the findings

Impact and novelty not assessed. Negative/inconclusive results accepted. Meaningful replication encouraged where rationale & benefit to literature is clearly stated.
As stated previously and explained in the text of the manuscript, the authors studied a high altitude Konservat-Lagerstätten which is rare, making their ecological and taphonomic study relevant to the field. However, the most interesting observation (the preservation of articulated frogs within microbial mats) has to be revisited, better explained and supported with more data.

All underlying data have been provided; they are robust, statistically sound, & controlled.
It is difficult to address the robustness of the data presented. Although the authors state a number of times that these results are supported by “numerous frog specimens” (e.g. line 239), they should explicit the number of specimens studied for each member of the formation and present a statistical description of these specimens (N total, N articulated, N complete, fragmentation, etc). This information will help to understand the magnitude of their work.

Conclusions are well stated, linked to original research question & limited to supporting results.
Conclusions are aseptic and contained, based on the observation exposed in the previous sections.

Speculation is welcome, but should be identified as such.
Although most of the paper is merely a description of their observations, there is a speculative section aimed to explain the first of the modes of preservation (lines 243-262). Despite speculation is frequently necessary for the understanding of this kind of complex processes, it has to be consistent and well supported by previous information.
It is reasonable to infer the presence of microbial mats based on these crenulated laminations but more supporting data are more than welcome. However, anoxic conditions and carbonate precipitation on mats generates a few concerns:
- First, in Iniesto et al. we have investigated the preservation of frogs in mats and preservation in mats using microelectrodes and such anoxic conditions were not observed. If the authors have data supporting these anoxic conditions they should present it or, in absence of these data, stick to the literature. I am fully aware of the use of “anoxic conditions” is frequent in taphonomy to explain delayed decay, but it has to be well supported.
- Subsequently, they discard the effect of anoxia based on the record observation (if you don’t have data supporting anoxia and the presence of a number of invertebrates can tell the opposite, why are you using this argument?) and they linked the absence of scavenging to carbonate precipitation. However, this carbonate precipitation can occur in two different modes. First, microbial mats are known to generate carbonates (Ludwig et al 2005 Limnology and Oceanography 50(6):1836-1843 or Kaźmierczak 2015 Life 5(1): 744–769) but they are not lithified structures. Consequently, scavengers would have complete access to frog corpses. In fact, the precipitation rate presented by the authors does not relate to any observation on mats and is merely experimental. Actual observations on freshwater microbialites (Brady 2009 Geobiology 7(5):544-55) estimated a microbialite growth rate of 0.05 mm per year (73 times less than what authors said). And this applies to microbialites and not to microbial mats (second mode of precipitation). If carbonate precipitation is dominant and really active (as stated by the authors), microbial communities can form lithified microbial mats known as microbialites. In that context frogs would be laying over an actual rock at the bottom of the lake and their burial will be then very difficult. The authors should not use microbial mats as a Deus ex machina being soft and smooth and able to cover frogs quickly and then mineralized and hard to avoid scavenging just after burial.

Additional comments

This study has the potential to be a useful addition to palaeoecological literature. The documentation of exceptional preservation is this high altitude formation is abundant and can be useful for further studies in taphonomy, ecology and evolution in this kind of formations. However, major aspects have to be amended prior to consider it publication as it stands.

Reviewer 2 ·

Basic reporting

EXCEPTIONAL PRESERVATION OF ?LATEST CRETACEOUS (MAASTRICHTIAN)-PALEOCENE FROGS (EORUBETA NEVADENSIS) OF THE SHEEP PASS FORMATION OF EAST-CENTRAL NEVADA AND IMPLICATIONS FOR PALEOGEOGRAPHY OF THE NEVADAPLANO

This paper is focused on the different states of preservation of specimens of a fossil frog discovered In the type section of the Sheep Pass Formation and on this basis makes some paleoecological inferences.

Title: It puts too much emphasis on the “exceptional” preservation, a characterization that usually has a different meaning in the literature (e.g., Poyato Ariza & Buscalioni, 2016). In fact, the specimens shown in the pictures are not exceptionally well preserved. It should better say something like “Preservation aspects of the latest Cretaceous………”, better reflecting the topics dealt with in the paper.
42 plant remains
48 ..and invertebrates such as mollusks…
61-62 Excellent preservation involves integrity, anatomic connection, and osteological detail. You are not showing a fully articulated specimen in the figure.
64 that represents …
103 a diverse mammalian assemblage, including also lizards, snakes, and anurans.

147 a diverse assemblage of plants, etc.

149. the term edentulous is ambiguous (teeth are not preserved?)

248. Microbial mat development may also have been responsible for maintaining articulation of the frog skeletons for prolonged periods. Articulation of skeletons of frogs remained unbroken after 24 months in experiments (see.g., Iniesto et al., 2016, Lopez-Garcia et al., 2016)). This aspect was not thoroughly explored.


Figure 7: This is an incomplete skeleton partially disarticulated (e.g., clavicles are separated from scapulae and coracoids, skull disarticulated from axial column, etc)

Figure 8: Most skeletal elements are disarticulated, although they remain near their natural positions.


Iniesto, M., Buscalioni, AD., Guerrero M.C., Benzerara, K., Moreira, D., Lopez-Archilla, A.I., 2016. Involvement of microbial mats in early fossilization by decay delay and formation of impressions and replicas of vertebrates and invertebrates. Scientific Reports 6:25716 | DOI: 10.1038/srep25716


Lopez-Garcia, A., Martin-Abad, H., Cambra-Moo, O., 2016. Anuran biostratinomy. In: Poyato-Ariza, F.J., Buscalioni, A.D., (editors), Las Hoyas: A Cretaceous wetland, pp. 211-215.
Poyato-Ariza, F.J., Buscalioni, A.D., 2016. Exceptional preservation. In: Poyato-Ariza, F.J., Buscalioni, A.D., (editors), Las Hoyas: A Cretaceous wetland, pp. 229- 230.

Experimental design

No comment

Validity of the findings

No comment

·

Basic reporting

no comment

Experimental design

no comment

Validity of the findings

no comment

Additional comments

This is an interesting manuscript describing a rather unique locality that is set apart by both its paleogeography (high altitude lake) and taphonomy (various frog horizons/preservational modes). I found the manuscript well organized and clearly written, the science sound and the conclusions very reasonable. I do have a few minor recommendations and I would hope that the corresponding changes would improve the effectiveness of the paper. These are outlined below followed by simple editing corrections. Given that I have no criticism of the underlying science, I would recommend publication with minor changes.

Taphonomic modes: There should be consistent treatment of these. They are numbered incompletely in the abstract, but not in the main body of the text. I would recommend that the five modes be numbered 1 thru 5 in the text and abstract. Also, the authors should include a table that lists the 5 modes, tapho description, their associated lithologies, member (B or C), and interpretation. This would be a helpful guide for the reader, and then useful for people potentially applying these modes elsewhere.

Better photographs: Could better pictures be provided for Figures 3, 5 and 8? Figure 8 seems to be somewhat unfocused, and the objects in 3 and 5 could be more visible. For figure 10, rather than showing two sides of the same cut surface, could one side be replaced by a close-up showing the ostracod lining? In Figure 12, could the contrast be enhanced so that the bones stood out a bit more?

Timing: The Sheep Pass Formation spans a very long time and the K/Pg. Please add a paragraph under Geologic Setting that discusses the age or potential age of Members B and C. I realize that there is likely uncertainty here, but it would be helpful for the reader to know what is known or unknown in terms of age. Specifically, do B and C and their frogs post date the K/Pg or span the K/Pg? Could any horizons reflect the K/Pg or can this all be confidently eliminated?

[Note: numbers are line numbers.]

64 - Reword “ an exception is“ to ‘except for’.

87- Omit “estimated to be”.

97 - Should be ‘Winfrey, 1958, 1960;’

99 - Replace “While” with ‘Whereas’.

101 - Omit “Sheep Pass”.

108-109 - Can you say something about Eorubeta? To what family or clade does it belong? Does this say anything about its ecology? Might this be relevant to the interpretation of the taphonomy?

120-121. Please explain these abbreviations here, when first used.

137-138. Here degree of association and degree of disassociation is recorded? Isnt’ this redundant? If so, simplify text. If not, explain what the difference is.

139. Light micriscope…Might there be something to be seen using a different light source to examine these specimens, e.g. UV? I’m not suggesting this is needed here, but perhaps moving forward with specimens.

178. Can you say anything about the plants relative to past palynology work (e.g., Fouch 1979)?

214-220. Doesn’t this section on decopods belong earlier with lines 189-196?

244. The wording here is confusing in terms of the dolomitic mudstones. So rather than, “within the crenulated-fabric, dolomitic mudstones”, revise to ‘only within those dolomitic mudstones with a crenulated fabric.’ Otherwise it sounds that all dm have crenularted fabric.

247-248. I didn’t understand the meaning nor context of this sentence. “This interpretation is supported by the stratigraphic distribution of specimens that had settled onto microbial mats in life position and were subsequently buried and preserved.” Can you help me out?

259. Make new sentence beginning with “Modern rates…..”

268. Can you describe a little more of what you mean by “more out of “life position”’?

296. Pers. communication - There are published references on crayfish that might be better here, although I don’t know if they discuss ostracods in the wall.

Hasiotis, S.T., Mitchell, C.E., Dubiel, R.F., 1993b. Application of morphologic burrow interpretations to discern continental bur- row architects: lungfish or crayfish. Ichnos 2, 315–333.
Hasiotis, S.T., Kirkland, J.I., 1997. Crayfish fossils and burrows from the Upper Jurassic Morrison Formation, Colorado Plateau, USA: implications for crayfish evolution. Freshwater Crayfish 11, 106–120.
Hasiotis, S.T., Mitchell, C.E., 1993. A comparison of crayfish bur- row morphologies: Triassic and Holocene fossil, paleo- and neo- ichnological evidence, and the identification of their burrowing signatures. Ichnos 2, 291–314.
333. Could there be other causes: anoxic events, water turnover, etc?
Genera; Q: Is there any size variation in the frogs between the various taphonomic modes?
References: Fiorillo 1989 and Eberth et al. 2006 are listed, but I did not find them in the text. These should be removed.

Figures. See also comments above.

Figure 5. Some labelling is needed here to know what is being referred to. Also, the word ‘left’ (?) is missing in the caption.

Figure 8. This image is just too blurry.

Figure 9. Could this be the trace Cochlichnus?

Figure 10. A close-up of the wall could one side of the burrow slice.

Figure 12 caption refers to a Figure 13. Either omit or add figure 13.

---

## Round 0.2 · accepted · Accept

Reviewer 3 suggests a few edits, which can be done while your manuscript is in Production.

Congratulations again!

·

Basic reporting

no comment

Experimental design

no comment

Validity of the findings

no comment

Additional comments

Revisions look good and have addressed all my concerns.

A few minor typos/edits:
75 - which
185 - Could simply say 'toothless mammal jaw' or 'edentulous mammal jaw'.
202 - Fourth not Forth.
301 - as not ais.
418 - Space needed between 'in' and 'Ontario'.
448-449 - 'those in Member C....events.' is redundant with next sentence. Should omit and combine two sentences.